# A Distributional Robustness Certificate by Randomized Smoothing

## Abstract

The robustness of deep neural networks against adversarial example attacks has received much attention recently. We focus on certified robustness of smoothed classifiers in this work, and propose to use the worst-case population loss over noisy inputs as a robustness metric. Under this metric, we provide a tractable upper bound serving as a robustness certificate by exploiting the duality. To improve the robustness, we further propose a noisy adversarial learning procedure to minimize the upper bound following the robust optimization framework. The smoothness of the loss function ensures the problem easy to optimize even for non-smooth neural networks. We show how our robustness certificate compares with others and the improvement over previous works. Experiments on a variety of datasets and models verify that in terms of empirical accuracies, our approach exceeds the state-of-the-art certified/heuristic methods in defending adversarial examples.

## 1 Introduction

Deep neural networks (DNNs) have been known to be vulnerable to adversarial example attacks: by feeding the DNN with slightly perturbed inputs, the attack alters the prediction output. The attack can be fatal in performance-critical systems such as autonomous vehicles or automated tumor diagnosis. A DNN is *robust* when it can resist such an attack that, as long as the range of the perturbation is not too large (usually invisible by human), the model produces an expected output despite of the specific perturbation. Various approaches have been proposed for improving the robustness of DNNs, with or without a performance guarantee.

Although a number of approaches have been proposed for certified robustness, it is vague how robustness should be defined. For example, works including Cohen et al. (2019); Pinot et al. (2019); Li et al. (2019); Lecuyer et al. (2019) propose smoothed classifiers to ensure the inputs with adversarial perturbation to be classified into the same class as the inputs without. However, since both inputs are inserted randomized noise, it cannot be guaranteed that the inputs are classified into the *correct* class. It is possible that the adversarially perturbed input has the same label as the original one which is wrongly classified by the DNN. In this case, the robustness guarantee does not make sense any more. Further, the robustness guarantee is provided at the instance level, *i.e.,* within a certain perturbation range, the modification of an input instance cannot affect the prediction output. But a DNN is a statistical model to be evaluated on the input distribution, rather than a single instance. Instead of counting the number of input instances meeting the robustness definition, it is desired to evaluate the robustness of a DNN over the input distribution.

We introduce the distributional risk as a DNN robustness metric, and propose a noisy adversarial learning (NAL) procedure based on distributional robust optimization, which provides a provable guarantee. Assume a base classifier $f$ trying to map instance $x_0$ to corresponding label $y$. It is found that when fed with the perturbed instance $x$ (within a $l_2$ ball centered at $x_0$), a smoothed classifier $g(x) = \mathbb{E}_Z[f(x+z)]$ with $z \sim Z = \mathcal{N}(0, \sigma^2 I)$ can provably return the same label as $g(x_0)$ does. However, we think such a robustness guarantee cannot ensure $g(x_0)$ to be correctly classified as $y$, resulting in unsatisfying performance in practice. Instead, we evaluate robustness as the worst-case loss over the distribution of noisy inputs. For simplicity, we jointly express the input instance and the label as $x_0 \sim P_0$ where $P_0$ is the distribution of the original input. By using $\ell(\cdot)$ as the loss function, we evaluate DNNs by the worst-case distributional risk: $\sup_S \mathbb{E}_S[\ell(\theta; s)]$. The classifier is

parameterized by $\theta \in \Theta$, and $s = x + z \sim S$ where $S$ is a distribution within a certain distance from $P_0$. We prove such a loss is upper bounded by a data-dependent certificate, which can be optimized by the noisy adversarial training procedure:

$$\underset{\theta \in \Theta}{\text{minimize}} \sup_S \mathbb{E}_S[\ell(\theta; s)]. \tag{1}$$

Compared to previous robustness certificates via smoothed classifiers, our method provides a provable guarantee w.r.t. the ground truth input distribution. Letting the optimized $\theta$ be the parameter of $g(\cdot)$ and $f(\cdot)$ respectively, we further show that the smoothed classifier $g(\cdot)$ provides an improved robustness certificate than that of $f(\cdot)$, due to a tighter bound on the worst-case loss.

The key is that, for mild perturbations, we adopt a Lagrangian relaxation for the usual loss $\ell(\theta; x+z)$ as the robust surrogate, and the surrogate is strongly concave in $x$ and hence easy to optimize. Our approach enjoys convergence guarantee similar to the method in Sinha et al. (2018), but different from Sinha et al. (2018), our approach does not require $\ell$ to be smooth, and thus can be applied to arbitrary neural networks. The advantage of the smoothed classifier also lies in a tighter robustness certificate than the base classifier. The intuition is that, in the inner maximization step, instead of seeking one direction which maximizes the loss, our approach performs gradient ascent along the direction which maximizes the total loss of examples sampled from the neighborhood of the original input. The noisy adversarial training procedure produces smoothed classifiers robust against the neighborhood of the worst-case adversarial examples with a certified bound.

Highlights of our contribution are as follows. *First,* we review the drawbacks in the previous definition of robustness, and propose to evaluate robustness by the worst-case loss over the input distribution. *Second,* we derive a data-dependent upper bound for the worst-case loss, constituting a robustness certificate. *Third,* by minimizing the robustness certificate in the training loop, we propose noisy adversarial learning for enhancing model robustness, in which the smoothness property entails the computational tractability of the certificate. Through both theoretical analysis and experimental results, we verify that our certified DNNs enjoy better accuracies compared with the state-of-the-art defending adversarial example attacks.

## 2   RELATED WORK

Works proposed to defend against adversarial example attacks can be categorized into the following categories.

In **empirical defences,** there is no guarantee how the DNN model would perform against the adversarial examples. Stability training (Zheng et al. (2016); Zantedeschi et al. (2017); Liu et al. (2018)) improves model robustness by adding randomized noise to the input during training but shows limited performance enhancement. Adversarial training (Goodfellow et al. (2015); Kurakin et al. (2018); Madry et al. (2017); Kannan et al. (2018); Zhang et al. (2019); He et al. (2019); Wang et al. (2019)) trains over adversarial examples found at each training step but unfortunately does not guarantee the performance over unseen adversarial inputs. Although without a guarantee, adversarial training has excellent performance in empirical defences against adversarial attacks.

**Certified defences** are certifiably robust against any adversarial input within an $\ell_p$-norm perturbation range from the original input. A line of works construct a computationally tractable relaxation for computing an upper bound on the worst-case loss over all valid attacks. The relaxations include linear programming (Wong & Kolter (2018)), mixed integer programming (Tjeng et al. (2018)), semidefinite programming (Raghunathan et al. (2018)), and convex relaxation (Namkoong & Duchi (2017); Salman et al. (2019b)). But those deterministic methods are not scalable. Some works such as Dvijotham et al. (2018) formulate the search for the largest perturbation range as an optimization problem and solve its dual problem. Sinha et al. (2018) also propose a robustness certificate based on a Lagrangian relaxation of the loss function, and it is provably robust against adversarial input distributions within a Wasserstein ball centered around the original input distribution. The certificate of our work is constructed on a Lagrangian relaxation form of the worst-case loss, but has a broader applicability than Sinha et al. (2018) with a tighter loss bound due to the smoothness property.

An alternative line of works propose to select appropriate surrogates for each neuron activation layer by layer (Weng et al. (2018); Zhang et al. (2018)) to facilitate the search for a certified lower bound. By integrating with interval bound propagation (Gowal et al. (2018)), Zhang et al. (2020)

make the search computationally efficient and scalable. Other works (Mirman et al. (2018); Singh et al. (2018)) apply the abstract interpretation to train provably robust neural networks. Our work is orthogonal to these works.

**Randomized smoothing** introduces randomized noise to the neural network, and tries to provide a statistically certified robustness guarantee. Pinot et al. (2020) have demonstrated by game theory that no deterministic classifier can claim to be more robust than all others against any possible adversarial attack. But such a question remains open in the randomized regime, where randomized smoothing can be considered as a contributing effort. The smoothing method does not depend on a specific neural network, or a type of relaxation, but can be generally applied to arbitrary neural networks. The idea of adding randomized noise was first proposed by Lecuyer et al. (2019), given the inspiration of the differential privacy property, and then Li et al. (2019) improve the certificate with Rényi divergence. Cohen et al. (2019) obtain a larger certified robustness bound through the smoothed classifier based on Neyman-Pearson theorem. Phan et al. (2020) extend the noise addition mechanism to large-scale parallel algorithms. By extending the randomized noise to the general family of exponential distributions, Pinot et al. (2019) unify previous approaches to preserve robustness to adversarial attacks. Lee et al. (2019) offer adversarial robustness guarantees for $\ell_0$-norm attacks. Both Salman et al. (2019a); Jia et al. (2019) employ adversarial training to improve the performance of randomized smoothing. Following a similar principle, our work trains over adversarial data with randomized noise. But we provide a more practical robustness certificate and a training method achieving higher empirical accuracy than theirs.

## 3    PROPOSED APPROACH

We first define the closeness between distributions, based on which we constrain how far the input distribution is perturbed. Then we introduce our definition of robustness on smoothed classifiers. Our main theorem gives a tractable robustness certificate which is easy to optimize. Our algorithm for improving the robustness of the smoothed classifiers is provided. All proofs are collected in the appendices for conciseness.

### 3.1    A DISTRIBUTIONAL ROBUSTNESS CERTIFICATE

**Definition 1** (Wasserstein distance). *Wasserstein distances define a notion of closeness between distributions. Let $\left(\mathcal{X} \subset \mathbb{R}^d, \mathcal{A}, P\right)$ be a probability space and the transportation cost $c : \mathcal{X} \times \mathcal{X} \to [0, \infty)$ be nonnegative, lower semi-continuous, and $c(x, x) = 0$. $P$ and $Q$ are two probability measures supported on $\mathcal{X}$. Let $\Pi(P, Q)$ denotes the collection of all measures on $\mathcal{X} \times \mathcal{X}$ with marginals $P$ and $Q$ on the first and second factors respectively,* i.e., *it holds that $\pi(A, \mathcal{X}) = P(A)$ and $\pi(\mathcal{X}, A) = Q(A)$, $\forall A \in \mathcal{A}$ and $\pi \in \Pi(P, Q)$. The Wasserstein distance between $P$ and $Q$ is*

$$W_c(P, Q) := \inf_{\pi \in \Pi(P,Q)} \mathbb{E}_\pi \left[ c\left(x, y\right) \right]. \tag{2}$$

For example, the $\ell_2$-norm $c(x, x_0) = \|x - x_0\|_2^2$ satisfies the aforementioned conditions.

**Distributional robustness.** Assume the original input $x_0$ is drawn from the distribution $P_0$, and the perturbed input $x$ is drawn from the distribution $P$. Each input is added randomized Gaussian noise $z \sim Z = \mathcal{N}(0, \sigma^2 I)$ before being fed to the classifier. Instead of regarding the noise as a part of the smoothed classifier, we treat $s = x + z$ as a noisy input coming from the distribution $S$ in the analysis. Since $z \in \mathbb{R}^d$, we need to set $\mathcal{X} = \mathbb{R}^d$ to admit $s \in \mathcal{X}$ as Lecuyer et al. (2019); Cohen et al. (2019); Salman et al. (2019a) do. Since the perturbed input should be visually indistinguishable from the original one, we define the robustness region as $\mathcal{P} = \{P : W_c(P, P_0) \leq \rho, P \in P(\mathcal{X})\}$, where $\rho > 0$. Within such a region, we evaluate the robustness as a worst-case population loss over noisy inputs: $\sup_{S \in \mathcal{P}} \mathbb{E}_S[\ell(\theta; s)]$. Essentially, we evaluate the robustness of a smoothed classifer based on its performance on the worst-case adversarial example distribution. A smaller loss indicates a higher level of robustness. We will compare the definition against others in the next section. However, such a robustness metric is impossible to measure in practice as we have no idea about $P$. Even if $P$ can be acquired, it can be a non-convex region which renders the constrained optimization objective intractable. Hence we resort to the Lagrangian relaxation of the problem by assuming a dual variable $\gamma$.

As the main theorem of this work, we provide an upper bound for the worst-case population loss for any level of robustness $\rho$. We further show that for small enough $\rho$, the upper bound is tractable and easy to optimize.

**Theorem 1.** *Let $\ell : \Theta \times \mathcal{X} \to \mathbb{R}$ and transportation cost function $c : \mathcal{X} \times \mathcal{X} \to \mathbb{R}_+$ be continuous. Let $x_0$ be an input drawn from the input distribution $P_0$, $x$ be the adversarial example which follows the distribution $P$ and $z \sim Z = \mathcal{N}(0, \sigma^2 I)$ be the additive noise of the same shape as $x$. The sum of $x$ and $z$ is denoted as $s = x + z \sim S$ and we let $\phi_\gamma(\theta; x_0) = \sup_{x \in \mathcal{X}} \mathbb{E}_Z \{\ell(\theta; x + z) - \gamma c(x + z, x_0)\}$ be the robust surrogate. For any $\gamma, \rho > 0$ and $\sigma$, we have*

$$\sup_{S: W_c(S, P_0) \leq \rho} \mathbb{E}_S[\ell(\theta; s)] \leq \gamma\rho + \mathbb{E}_{P_0}[\phi_\gamma(\theta; x_0)]. \tag{3}$$

The proof is given in Appendix A.1. It is notable that the right-hand side take the expectation over $P_0$ and $Z$ respectively, and given a particular input $x_0$ and a noise sample $z$, we seek an adversarial example which maximizes the surrogate loss. Typically, $P_0$ is impossible to obtain and thus we use an empirical distribution, such as the training data distribution, to approximate $P_0$ in practice.

Since Thm. 1 provides an upper bound for the worst-case population loss, it offers a principled adversarial training approach which minimizes the upper bound instead of the actual loss, *i.e.,*

$$\underset{\theta \in \Theta}{\text{minimize}} \ \mathbb{E}_{P_0}[\phi_\gamma(\theta; x_0)]. \tag{4}$$

In the following we show the above loss function has a form which is tractable for arbitrary neural networks, due to a smoothed loss function. Hence Thm. 1 provides a tractable robustness certificate depending on the data.

**Properties of the smoothed classifier.** We show the optimization objective of Eq. 4 has a form which is tractable for any neural network, particular for the non-smooth ones with ReLU activation layers. More importantly, the smoothness of the classifier enables the adversarial training procedure to converge as we want by using the common optimization techniques such as stochastic gradient descent. The smoothness of the loss function comes from the smoothed classifier with randomized noise. Specifically,

**Theorem 2.** *Assume $\ell : \Theta \times \mathcal{X} \to [0, M]$ is a bounded loss function. The loss function on the smoothed classifier can be expressed as $\hat{\ell}(\theta; x) := \mathbb{E}_Z[\ell(\theta; x + z)]$, $z \sim Z = \mathcal{N}(0, \sigma^2 I)$. Then we have $\hat{\ell}$ is $\frac{2M}{\sigma^2}$-smooth w.r.t. $\ell_2$-norm, i.e., $\hat{\ell}$ satisfies*

$$\left\|\nabla_x \hat{\ell}(\theta; x) - \nabla_x \hat{\ell}(\theta; x')\right\|_2 \leq \frac{2M}{\sigma^2}\left\|x - x'\right\|_2. \tag{5}$$

The proof is in Appendix A.2. It mainly takes advantage of the randomized noise which has a smoothing effect on the loss function. For DNNs with non-smooth layers, the smoothed classifier makes it up and turns the loss function to a smoothed one, which contributes as an important property to the strong concavity of $\mathbb{E}_Z[\ell(\theta; x + z) - \gamma c(x + z, x_0)]$ and therefore ensures the tractability of the robustness certificate.

**Corollary 1.** *For any $c : \mathcal{X} \times \mathcal{X} \to \mathbb{R}_+ \cup \{\infty\}$ 1-strongly convex in its first argument, and $\hat{\ell} : x \mapsto \mathbb{E}_Z[\ell(\theta; x + z)]$ being $\frac{2M}{\sigma^2}$-smooth, the function $\mathbb{E}_Z\{\ell(\theta; x + z) - \gamma c(x + z, x_0)\}$ is strongly concave in $x$ for any $\gamma \geq \frac{2M}{\sigma^2}$.*

The proof is in Appendix A.3. Note that here we specify the requirement on the transportation cost $c$ to be 1-strongly convex in its first argument. The $\ell_2$-norm cost satisfies the condition. Before showing how the strong concavity plays a part in the convergence, we illustrate our algorithm first.

## 3.2 NOISY ADVERSARIAL LEARNING ALGORITHM

Problem 4 provides an explicit way to improve the robustness of a smoothed classifier parameterized by $\theta$. We correspondingly design a noisy adversarial learning algorithm to obtain the classifier of which its robustness can be guaranteed. In the algorithm, we use the empirical distribution to replace the ideal input distribution $P_0$, and sample $z$ a number of times to substitute the expectation with

the sample average. Assuming we have a total of $n$ training instances $x_0^i, \forall i \in [n]$, and sample $z_{ij} \sim \mathcal{N}(0, \sigma^2 I)$ for the $i$-th instance for $r$ times, the objective is:

$$\underset{\theta \in \Theta}{\text{minimize}} \ \frac{1}{nr} \sum_{i=1}^{n} \sum_{j=1}^{r} \sup_{x \in \mathcal{X}} \left[ \ell(\theta; x + z_{ij}) - \gamma c \left( x + z_{ij}, x_0^i \right) \right]. \tag{6}$$

The detail of the algorithm is illustrated in Alg. 1. In the inner maximization step (line 3-6), we adopt the *projected gradient descent* (PGD Madry et al. (2017); Kurakin et al. (2018)) to approximate the maximizer according to the convention. The hyperparameters include the number of iterations $K$ and the learning rate $\eta_1$. Within each iteration, we sample the Gaussian noise $r$ times, given which we compute an average perturbation direction for each update. The more noise samples, the closer the averaging result is to the expectation value, which is definitely at the sacrifice of higher computation expense. Similarly, a larger number of $K$ indicates stronger adversarial attacks and higher model robustness, but also incurs higher computation complexity. Hence choosing appropriate values of $r$ and $K$ is important in practice.

---

**Algorithm 1** Training Phase of NAL

---

**Input:** batch size $n$, number of noise samples $r$, noise STD $\sigma$, learning rate $\eta_1, \eta_2$, number of iterations $K$, penalty parameter $\gamma$, training iterations $T$
**Output:** the classifier parameter $\theta$
1: **for** $t \in \{1, \dots, T\}$ **do**
2:     **for** $i \in \{1, \dots, n\}$ **do**
3:         **for** $k \in \{0, \dots, K-1\}$ **do**
4:             $\Delta x_k^i = \frac{1}{r} \sum_{j=1}^{r} \nabla_{x_k^i} \ell(\theta; x_k^i + z_{ij}) - \gamma \nabla_{x_k^i} c(x_k^i + z_{ij}, x_0^i)$, where $z_{ij} \sim \mathcal{N}(0, \sigma^2 I)$
5:             $x_{k+1}^i = x_k^i + \eta_1 \Delta x_k^i$
6:         **end for**
7:     **end for**
8:     $\theta^{t+1} = \theta^t - \eta_2 \left\{ \frac{1}{nr} \sum_{i=1}^{n} \left[ \nabla_\theta \sum_{j=1}^{r} \ell(\theta^t; x_K^i + z_{ij}) \right] \right\}$
9: **end for**

---

After training is done, we obtain the classifier parameter $\theta$. In the inference phase, we sample a number of $z \sim \mathcal{N}(0, \sigma^2 I)$ to add to the testing instance. The noisy testing examples are fed to the classifier to get the prediction outputs.

**Convergence.** An important property associated with the smoothed classifier is the strong concavity of the robust surrogate loss, which is the key to the convergence proof. The detail of the proof can be found in Appendix A.4. As long as the loss $\hat{\ell}$ is smooth on the parameter space $\Theta$, NAL has a convergence rate $O(1/\sqrt{T})$, similar to Sinha et al. (2018), but NAL does not need to replace the non-smooth layer ReLU with Sigmoid or ELU to guarantee robustness.

## 4 A Tighter Bound

We compare our work with the state-of-the-art robustness definitions and certificates in this section.

### 4.1 Adversarial Training

Our approach improves the distributional robustness certificate proposed by Sinha et al. (2018). In Sinha et al. (2018), a classifier $f$ maps input instance $x_0 \sim P_0$ to corresponding label $y$. They perturb $x_0$ to $x'$ in the same robustness region as ours: $\mathcal{P} = \{P : W_c(P, P_0) \leq \rho, P \in P(\mathcal{X})\}$, where $\rho > 0$. But their worst-case population loss is defined on the base classifier without noise: $\sup_{P':W_c(P',P_0) \leq \rho} \mathbb{E}_{P'}[\ell(\theta; x')]$. We show that, given the same classifier parameter $\theta$, our worst-case loss is smaller than Sinha et al. (2018), suggesting a better robustness certificate.

**Theorem 3.** *Under the same denotations and conditions as Thm. 1, we have*

$$\sup_{S \in \mathcal{P}} \mathbb{E}_S[\ell(\theta; s)] \leq \inf_{\gamma \geq 0} \{ \gamma \rho + \mathbb{E}_{P_0}[\phi_\gamma(\theta; x_0)] \}$$

$$\leq \inf_{\gamma \geq 0} \left\{ \gamma \rho + \mathbb{E}_{P_0} \sup_{x' \in \mathcal{X}} [\ell(\theta; x') - \gamma c(x', x_0)] \right\} = \sup_{P':W_c(P',P_0) \leq \rho} \mathbb{E}_{P'}[\ell(\theta; x')]. \tag{7}$$

The proof is given in Appendix A.5. We demonstrate that not only the worst-case loss is smaller, but the tractable upper bound is smaller than the certificate of Sinha et al. (2018). If the outer minimization problem applies to both sides of the inequality, our approach would obtain a smaller loss when both classifiers share the same neural architecture.

## 4.2 SMOOTHED CLASSIFIERS

Works including Lecuyer et al. (2019); Cohen et al. (2019); Pinot et al. (2019); Li et al. (2019) and others guarantee the robustness of a DNN classifier by inserting randomized noise to the input at the inference phase. Most of them do not concern about the training phase, but merely provide a deterministic relationship between the robustness certificate and the additive noise. Specifically, we have the original input $x_0 \in \mathcal{X}$ and its perturbation $x$ within a given range $\|x - x_0\|_2 \leq \varepsilon$. The smoothed classifier $g(x)$ returns class $c_i$ with probability $p_i$. For instance $x_0$, robustness is defined by the largest perturbation radius $R$ which does not alter the instance's prediction, *i.e.,* $g(x)$ is classified into the same category as $g(x_0)$. Such perturbation radius depends on the largest and second largest probabilities of $p_i$, denoted by $p_A, p_B$ respectively. For example, the results in Cohen et al. (2019) have shown that $R = \frac{\sigma}{2}\left(\Phi^{-1}\left(\underline{p_A}\right) - \Phi^{-1}\left(\overline{p_B}\right)\right)$ where $\Phi^{-1}$ is the inverse of the standard Gaussian CDF, $\underline{p_A}$ is a lower bound of $p_A$, and $\overline{p_B}$ is an upper bound of $p_B$.

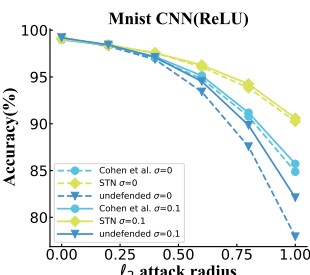

**Figure 1:** Accuracies of models trained on MNIST under different levels of $\ell_2$ attacks. Undefend means a naturally trained model. Solid lines represent models tested with additive noise, and dotted lines mean that without. $\sigma = 0.1$ means adding Gaussian noise $\mathcal{N}(0, 0.1^2 \boldsymbol{I})$.

The previous robustness definition only guarantees $g(x)$ to be classified to the same class as $g(x_0)$, but ignores the fact that $g(x_0)$ may be wrongly classified, which is not a precise definition. To make up for it, Li et al. (2019) propose stability training with noise (STN) and Cohen et al. (2019) adopt training with noise, both of which learn smoothed classifiers mapping noisy inputs to correct labels. However, there is no guarantee to ensure $g(x_0)$ to be correctly labeled. Actually we found the robustness mainly comes from the STN/training with noise, rather than the noise addition at the inference. In Fig. 1, we could observe that the model performance indeed improves when tested with noise. However, the classifier trained without additive noise (triangle) degrades significantly compared with STN/training with noise (diamond/circle). The result is an evidence that a classifier almost cannot defend adversarial attacks when trained without but tested with additive noise. Therefore, we conclude the smoothed classifier can only improve robustness only if the base classifier is robust.

We consider robustness refers to the ability of a *DNN* to classify adversarial examples into the correct classes, and such an ability should be evaluated on the population of adversarial examples, not a single instance.

## 5 EXPERIMENT

**Baselines, datasets and models.** Testing accuracies under different levels of adversarial attacks are chosen as the metric. We compare the empirical performance of NAL with representative baselines including: WRM (Sinha et al. (2018)), SmoothAdv (Salman et al. (2019a)), STN (Li et al. (2019)) and TRADES (Zhang et al. (2019)). Since WRM requires the loss function to be smooth, we follow the convention to adapt the ReLU activation layer to the ELU layer. SmoothAdv combines adversarial training with the smoothed classifier and claims to be superior than Cohen et al. (2019). Hence we omit Cohen et al. (2019) in comparison. TRADES is an adversarial training algorithm which won 1st place in the NeurIPS 2018 Adversarial Vision Challenge. Experiments are conducted on datasets MNIST, CIFAR-10, and Tiny ImageNet, and models including a three-layer CNN, ResNet-18, VGG-16, and their corresponding variants with ReLu replaced by ELU for fair comparison with WRM. The cross-entropy loss is chosen for $\ell$ and $c(x, x_0) = \|x - x_0\|_2^2$ is selected as the cost function.

**Training hyperparameters.** Table 1 gives the training hyperparameters in NAL and the batch size is chosen as 128. The hyperparameters used in baselines are supplied in Appendix B.1. Since NAL

| Dataset | $\eta_1$ | $\eta_2$ | epochs | $\sigma$ | $\gamma$ | $\varepsilon$ |
|---|---|---|---|---|---|---|
| MNIST | $0.5/\gamma$ | $1 \times 10^{-4}$ | 25 | 0.05 | $\{0.25, 1.5, 3\}$ | $\{0.84, 0.34, 0.21\}$ |
| CIFAR-10(ResNet-18) | $0.5/\gamma$ | $1 \times 10^{-4}$ | 100 | 0.1 | $\{0.25, 1.5, 5\}$ | $\{1.53, 0.92, 0.40\}$ |
| CIFAR-10(VGG-16) | $0.5/\gamma$ | $1 \times 10^{-4}$ | 100 | 0.1 | $\{0.25, 1.5, 5\}$ | $\{1.23, 0.57, 0.28\}$ |
| Tiny ImageNet | $0.5/\gamma$ | $2 \times 10^{-5}$ | 100 | 0.1 | 1.5 | 0.93 |

**Table 1:** Hyperparameters and perturbation ranges on different datasets.

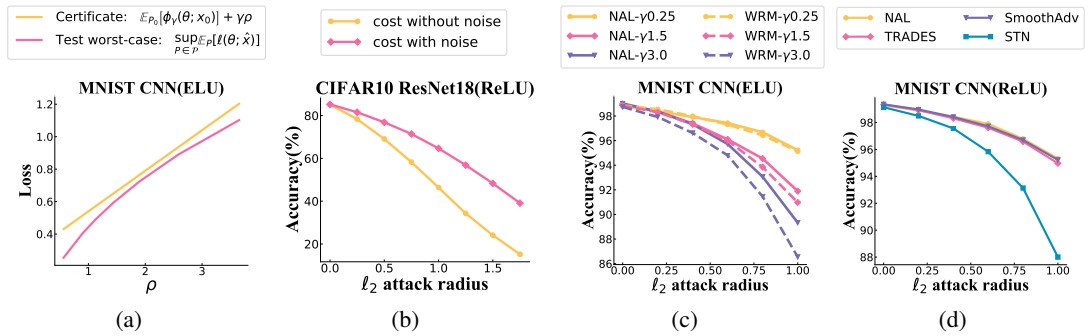

**Figure 2:** (a) gives the distance between the robustness certificate (yellow) and the worst-case performance on testing data (pink) with an example on MNIST. The gap between the two lines indicates the tightness of our certificate (Eq. 3). (b) compares the performance of two models trained with different $c(\cdot)$s. The classifier trained with the noise included in the cost has better performance overall. (c) compares the performance of NAL with WRM on MNIST, CNN (ELU) under different $\gamma$s. NAL overall has better performance than WRM. (d) compares NAL with SmoothAdv, TRADES and STN on MNIST, CNN at $\gamma = 0.25$ and the corresponding $\varepsilon$. NAL does not show significant improvement when $\gamma$ is small.

and WRM bound the adversarial perturbations by the Wasserstein distance $\rho$ which is different from the $\ell_2$-norm perturbation range $\varepsilon$ in SmoothAdv and TRADES, we need to establish an equivalence between the perturbation ranges in different methods. Following the convention of Sinha et al. (2018), we choose different $\gamma$s and for each $\gamma$ we generate adversarial examples $x$ by PGD with 15 iterations. We compute $\rho$ as the expected transportation cost between the generated adversarial examples and the original inputs over the training set:

$$\varepsilon^2 = \rho(\theta) = \mathbb{E}_{P_0}\mathbb{E}_Z\left[c\left(x + z, x_0\right)\right]. \tag{8}$$

And $\varepsilon$ can be computed accordingly. The corresponding values of $\gamma$ and $\varepsilon$ used in experiments are given in Table 1 as well.

**Attack parameters.** To evaluate the empirical accuracies for different methods, we adopt the PGD attack Kurakin et al. (2018); Madry et al. (2017) as the adversarial attack following the convention of Li et al. (2019); Sinha et al. (2018); Zhang et al. (2019), etc. We set the number of iterations in PGD attack as $K_{\text{attack}} = 20$ and the learning rate $\eta = 2\varepsilon_{\text{attack}}/K_{\text{attack}}$ where $\varepsilon_{\text{attack}}$ is $\ell_2$ attack radius.

## 5.1 RESULTS

**Certificate.** To better understand how close the upper bound is to the true distributional risk, we plot our certificate $\gamma\rho + \mathbb{E}_{\widehat{P}_{\text{test}}}\left[\phi_\gamma(\theta; x_0)\right]$ against any level of robustness $\rho$, and the out-of-sample (test) worst-case performance $\sup_{S \in \mathcal{P}} \mathbb{E}_S[\ell(\theta; s)]$ for NAL (Fig. 2(a)). Since the worst-case loss is hard to evaluate directly, we solve its Lagrangian relaxation for different values of $\gamma_{adv}$. For each $\gamma_{adv}$, we compute the average distance to adversarial examples in the test set as $\widehat{\rho}_{\text{test}}(\theta) := \mathbb{E}_{\widehat{P}_{\text{test}}} \mathbb{E}_Z\left[c\left(x_\star + z, x_0\right)\right]$ where $\widehat{P}_{\text{test}}$ is the test data distribution and $x_\star = \arg\max_x \mathbb{E}_Z\left\{\ell(\theta; x + z) - \gamma_{adv}c(x + z, x_0)\right\}$ is the adversarial perturbation of $x_0$. The worst-case loss is given by $\left(\widehat{\rho}_{\text{test}}(\theta), \mathbb{E}_{\widehat{P}_{\text{test}}} \mathbb{E}_Z\left[\ell(\theta; x_\star + z)\right]\right)$. As we observe, $\widehat{\rho}_{\text{test}}(\theta)$ tends to increase with a higher noise level. Hence we need to keep the noise at an appropriate level to make our certificate tractable.

**Cost without noise.** To find out if NAL works when noise is removed from the cost, we designed a verification experiment on CIFAR-10 (ResNet-18) by letting $c(x, x_0) = \|x - x_0\|_2^2$ and inserting noise only to $\ell$. We set $\gamma = 1.5, \sigma = 0.1, K = 4, r = 4$. As Fig. 2(b) has shown, the accuracy performance of the model excluding noise from the cost is far inferior, which shows that the randomized noise is an inherent part in the design.

**Sample number and PGD iterations.** We also study the impact of the noise sample number $s$ and PGD iteration $K$ to the model robustness with CIFAR-10 (ResNet-18) as an example. The result in

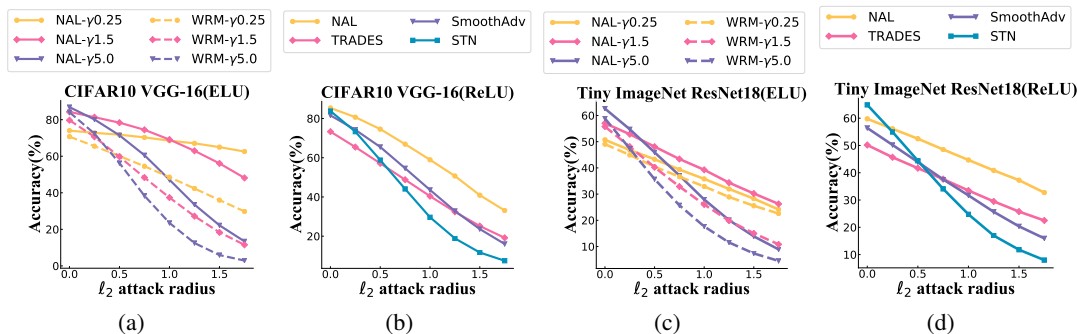

**Figure 3:** NAL outperforms baselines on CIFAR-10, VGG-16 and Tiny ImageNet, ResNet-18. (a),(c) are trained on ELU models under different $\gamma$s. For the same $\gamma$, NAL exceeds WRM. (b),(d) are trained on ReLU models with $\gamma = 1.5$ and the corresponding $\varepsilon$. NAL yields the highest robustness under different levels of attack. STN has the highest clean accuracy.

Table 2 shows that while the model performance enhances with $K$, it does not necessarily increase with a larger noise samples. We did not test with greater noise samples due to high complexity. For a combined consideration of computation overhead and accuracy, we choose $K = 4, r = 4$ by default in the experiments, which is likely to deliver a sufficiently good performance. Due to space constraints, complete experimental results are in Appendix B.2.

**Penalty and noise level.** We vary the value of $\gamma$ and $\sigma$ in the experiments to find out their impact. By the results in Fig. 2 (c),(d) and 3, we observe $\gamma = 0.25$ yields the best performance for MNIST, and $\gamma = 1.5$ is best for CIFAR-10 and Tiny ImageNet, considering all levels of adversarial attacks. For a complete result on $\gamma$, one can refer to Appendix B.3. Likewise, the best value of $\sigma$ also depends on the dataset, shown by the experimental results in Appendix B.2.

**Comparison with baselines.** Finally, we compare the empirical accuracies with the baselines and the results are presented in Fig. 2 (c),(d) and 3. For WRM, the experiments are conducted on the modified structure of DNNs to ensure smoothness. NAL has superior performance in almost all cases except that: 1) the clean accuracies (denoted by $\ell_2$ attack radius = 0) on CIFAR-10 and Tiny ImageNet of NAL are inferior to STN; 2) on MNIST, the performance of NAL is no worse but does not exceed baselines by a large margin. For 1), we found STN mostly has far worse performance than other schemes when the attack radius > 0, which echos the proposition in Salman et al. (2019a) that adversarial training brings higher robustness than stability training. Hence it can be explained by the inherent tradeoff between clean accuracy and robustness (Zhang et al. (2019)) that STN has higher clean accuracies than others. Actually, NAL shows better tradeoff between accuracy and robustness than baselines, indicated by the relatively flat accuracy lines. For 2), we think MNIST has a relatively simple decision boundary than the other two datasets and hence allows larger perturbations (smaller $\gamma$). Thus the performance boost by NAL is not significant. Actually, when $\gamma$ is larger, the performance of NAL exceeds baselines by a large margin (Appendix B.3).

| $\ell_2$ attack radius | 0 | 0.25 | 0.5 | 0.75 | 1 | 1.25 | 1.5 | 1.75 |
|---|---|---|---|---|---|---|---|---|
| $(K, r) = (4, 1)$ | **0.8647** | 0.7540 | 0.5950 | 0.4297 | 0.2814 | 0.1713 | 0.0983 | 0.0520 |
| $(K, r) = (4, 4)$ | 0.8546 | 0.7643 | 0.6520 | 0.5202 | 0.3922 | 0.2765 | 0.1811 | 0.1120 |
| $(K, r) = (4, 8)$ | 0.8537 | 0.7622 | 0.6482 | 0.5171 | 0.3846 | 0.2641 | 0.1702 | 0.0997 |
| $(K, r) = (8, 1)$ | 0.8593 | 0.7555 | 0.6091 | 0.4630 | 0.3260 | 0.2205 | 0.1453 | 0.0929 |
| $(K, r) = (8, 4)$ | 0.8517 | **0.7663** | **0.6566** | 0.5289 | 0.3970 | 0.2769 | 0.1827 | 0.1129 |
| $(K, r) = (8, 8)$ | 0.8493 | 0.7582 | 0.6520 | **0.5302** | **0.3978** | **0.2828** | **0.1895** | **0.1151** |

**Table 2:** Testing accuracies of NAL (CIFAR-10, ResNet-18) on a variety of $r$ and $K$. Under each setting, the model with the highest clean accuracy ($\ell_2$ attack radius = 0) is chosen for testing. Numbers in bold represent the best performance in defending the attack.

## 6 CONCLUSION

Our work view the robustness of a smoothed classifier from a different perspective, *i.e.,* the worst-case population loss over the input distribution. We provide a tractable upper bound (certificate) for the loss and devise a noisy adversarial learning approach to obtain a tight certificate. Compared with previous works, our certificate is practically meaningful and offers superior empirical robustness performance.

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

# A  PROOFS

## A.1  PROOF OF THEOREM 1

*Proof.* We express the worst-case loss in its dual form with dual variable $\gamma$. By the weak dual property, we have

$$\sup_{S \in \mathcal{P}} \mathbb{E}_S[\ell(\theta; s)] \leq \inf_{\gamma \geq 0} \sup_{S \in \mathcal{P}} \{\mathbb{E}_S[\ell(\theta; s)] - \gamma W_c(S, P_0) + \gamma \rho\}, \tag{9}$$

the left hand-side of which can be rewritten in integral form:

$$\inf_{\gamma \geq 0} \sup_{S \in \mathcal{P}} \left\{ \mathbb{E}_S[\ell(\theta; x + z)] - \gamma W_c(S, P_0) + \gamma \rho \right\}$$
$$= \inf_{\gamma \geq 0} \sup_{S \in \mathcal{P}} \left\{ \int \ell(\theta; x + z) dZ(z) P(x) - \gamma W_c(S, P_0) + \gamma \rho \right\}. \tag{10}$$

Note that for any $\pi \in \Pi(S, P_0)$, we have $\int f(s) dS = \iint f(s) d\pi(s, x_0)$. And by the definition of Wasserstein distance, we have

$$\inf_{\gamma \geq 0} \sup_{S \in \mathcal{P}} \left\{ \int \ell(\theta; s) dS(s) - \gamma W_c(S, P_0) + \gamma \rho \right\}$$
$$= \inf_{\gamma \geq 0} \sup_{S \in \mathcal{P}} \left\{ \iint \ell(\theta; s) d\pi(s, x_0) - \gamma \inf_{\pi \in \Pi(S, P_0)} \iint c(s, x_0) d\pi(s, x_0) + \gamma \rho \right\} \tag{11}$$
$$= \inf_{\gamma \geq 0} \sup_{S \in \mathcal{P}} \left\{ \sup_{\pi \in \Pi(S, P_0)} \iint [\ell(\theta; s) - \gamma c(s, x_0)] d\pi(s, x_0) + \gamma \rho \right\}.$$

By the independence between $z$ and $x, x_0$, one would obtain

$$\iint [\ell(\theta; s) - \gamma c(s, x_0)] d\pi(s, x_0) = \iiint [\ell(\theta; x + z) - \gamma c(x + z, x_0)] dZ(z) d\pi(x, x_0) \tag{12}$$

By taking the maximum over $x$,

$$\iiint [\ell(\theta; x + z) - \gamma c(x + z, x_0)] dZ(z) d\pi(x, x_0)$$
$$= \iint \mathbb{E}_Z[\ell(\theta; x + z) - \gamma c(x + z, x_0)] d\pi(x, x_0) \tag{13}$$
$$\leq \iint \sup_x \left\{ \mathbb{E}_Z[\ell(\theta; x + z) - \gamma c(x + z, x_0)] \right\} d\pi(x, x_0).$$

Fixing $x$ to be value that maximizes the expression to be integrated, $x$ in the formula is fixed, so we only need to integrate $d\pi(x, x_0)$ on $X$. So we can get:

$$\iint \sup_x \left\{ \mathbb{E}_Z[\ell(\theta; x + z) - \gamma c(x + z, x_0)] \right\} d\pi(x, x_0)$$
$$= \int_{x_0} \sup_x \left\{ \mathbb{E}_Z[\ell(\theta; x + z) - \gamma c(x + z, x_0)] \right\} dP_0(x_0) \tag{14}$$
$$= \mathbb{E}_{P_0} \sup_x \mathbb{E}_Z[\ell(\theta; x + z) - \gamma c(x + z, x_0)].$$

Because the distribution of $z$ is definite and $z$ is independent of $x$, and supremum of $S$ is replaced by the supremum of $x$. Therefore, Eq. 11 can be written as

$$\inf_{\gamma \geq 0} \sup_{S \in \mathcal{P}} \left\{ \sup_{\pi \in \Pi(S, P_0)} \iint [\ell(\theta; s) - \gamma c(s, x_0)] d\pi(s, x_0) + \gamma \rho \right\}$$
$$\leq \inf_{\gamma \geq 0} \sup_{S \in \mathcal{P}} \left\{ \sup_{\pi \in \Pi(S, P_0)} \mathbb{E}_{P_0} \sup_x \mathbb{E}_Z[\ell(\theta; x + z) - \gamma c(x + z, x_0)] + \gamma \rho \right\} \tag{15}$$
$$= \inf_{\gamma \geq 0} \left\{ \mathbb{E}_{P_0} \sup_x \mathbb{E}_Z[\ell(\theta; x + z) - \gamma c(x + z, x_0)] + \gamma \rho \right\}.$$

By plugging the above into Eq. 9, we could get

$$\sup_{S \in \mathcal{P}} \mathbb{E}_S[\ell(\theta; s)] \leq \inf_{\gamma \geq 0} \left\{ \mathbb{E}_{P_0} \sup_x \mathbb{E}_Z[\ell(\theta; x + z) - \gamma c(x + z, x_0)] + \gamma \rho \right\}$$
$$= \inf_{\gamma \geq 0} \left\{ \mathbb{E}_{P_0}[\phi_\gamma(\theta; x_0)] + \gamma \rho \right\} \leq \mathbb{E}_{P_0}[\phi_\gamma(\theta; x_0)] + \gamma \rho. \tag{16}$$

for any given $\gamma \geq 0$, which completes the proof. □

## A.2 Proof of Theorem 2

*Proof.* The proof of $\hat{\ell}$ being $\frac{2M}{\sigma^2}$-smooth is equivalent to $\nabla \hat{\ell}$ being $\frac{2M}{\sigma^2}$-*Lipschitz.* We apply the Taylor expansion in $\nabla \hat{\ell}$ at $x_0$ and set $\delta = x_0 - x$:

$$\nabla \hat{\ell}(x_0) = \nabla \hat{\ell}(x) + \nabla^2 \hat{\ell}(x + \theta \delta)\delta, \tag{17}$$

where $0 < \theta < 1$. Hence we only need to prove $\|\nabla^2 \hat{\ell}(x + \theta \delta)\|_2$ is bounded since $\|\nabla \hat{\ell}(x + \delta) - \nabla \hat{\ell}(x)\|_2 = \|\nabla^2 \hat{\ell}(x + \theta \delta)\delta\|_2$. By taking the first and second-order derivatives of $\hat{\ell}(x)$, we have

$$\nabla \hat{\ell}(x) = \frac{1}{(2\pi)^{d/2}\sigma^{d+2}} \int_{\mathbb{R}^d} \ell(t)(t - x) \exp\left(-\frac{1}{2\sigma^2}\|x - t\|^2\right) dt, \tag{18}$$

and

$$\nabla^2 \hat{\ell}(x) = \frac{1}{(2\pi)^{d/2}\sigma^{d+2}} \int_{\mathbb{R}^d} \ell(t) \exp\left(-\frac{1}{2\sigma^2}\|x - t\|^2\right)[-\boldsymbol{I} + \frac{1}{\sigma^2}(t - x)(t - x)^\top]dt. \tag{19}$$

We divide the right hand-side of Eq. 19 into two halves with the first half:

$$\|\frac{1}{(2\pi)^{d/2}\sigma^{d+2}} \int_{\mathbb{R}^d} \ell(t) \exp\left(-\frac{1}{2\sigma^2}\|x - t\|^2\right)(-\boldsymbol{I})dt\|_2$$
$$= \frac{1}{\sigma^2}\|\hat{\ell}(x)(-\boldsymbol{I})\|_2 \leq \frac{1}{\sigma^2}\|\hat{\ell}(x)\|_2 \leq \frac{M}{\sigma^2}. \tag{20}$$

The second half is

$$\|\frac{1}{(2\pi)^{d/2}\sigma^{d+4}} \int_{\mathbb{R}^d} \ell(t) \exp\left(-\frac{1}{2\sigma^2}\|x - t\|^2\right)((t - x)(t - x)^\top)dt\|_2$$
$$\leq \frac{1}{(2\pi)^{d/2}\sigma^{d+4}} \int_{\mathbb{R}^d} |\ell(t)| \exp\left(-\frac{1}{2\sigma^2}\|x - t\|^2\right)\|(t - x)(t - x)^\top\|_2 dt \tag{21}$$
$$\leq \frac{M}{(2\pi)^{d/2}\sigma^{d+4}} \int_{\mathbb{R}^d} \exp\left(-\frac{1}{2\sigma^2}\|x - t\|^2\right)\|(t - x)(t - x)^\top\|_2 dt.$$

Due to the rank of the matrix $(t - x)(t - x)^\top$ is 1, its $\ell_2$ norm is easy to compute:

$$\|(t - x)(t - x)^\top\|_2 = (t - x)^\top(t - x). \tag{22}$$

Hence

$$\frac{M}{(2\pi)^{d/2}\sigma^{d+4}} \int_{\mathbb{R}^d} \exp\left(-\frac{1}{2\sigma^2}\|x - t\|^2\right)\|(t - x)(t - x)^\top\|_2 dt = \frac{M}{\sigma^2}. \tag{23}$$

Finally, combining the two halves we get

$$\|\nabla^2 \hat{\ell}(x + \theta \delta)\|_2 \leq \frac{2M}{\sigma^2}. \tag{24}$$

□

## A.3 Proof of Corollary 1

*Proof.* Since $\hat{\ell}$ is $\frac{2M}{\sigma^2}$-smooth and $c$ is 1-strongly convex in its first argument, we have

$$\nabla_x^2 \hat{\ell}(\theta; x) \preceq \frac{2M}{\sigma^2}\boldsymbol{I}, \text{ and} \tag{25}$$

$$\nabla_x^2 \mathbb{E}_Z c(x + z, z_0) = \nabla_x^2 \left[c(x, x_0) + d\sigma^2\right] = \nabla_x^2 c(x, x_0) \succeq \boldsymbol{I}. \tag{26}$$

Therefore we have

$$\nabla_x^2 \mathbb{E}_Z \left\{\ell(\theta; x + z) - \gamma c(x + z, x_0)\right\} \preceq (\frac{2M}{\sigma^2} - \gamma)\boldsymbol{I}. \tag{27}$$

Hence the strong concavity is proved for $\gamma \geq \frac{2M}{\sigma^2}$. □

## A.4 CONVERGENCE PROOF

We start with the required assumptions, which roughly quantify the robustness we provide.

**Assumption 1.** *The loss $\hat{\ell} : \Theta \times \mathcal{X} \to [0, M]$ satisfies the Lipschitzian smoothness conditions*

$$
\begin{aligned}
\|\nabla_\theta \hat{\ell}(\theta; x) - \nabla_\theta \hat{\ell}(\theta'; x)\|_* &\leq L_{\theta\theta} \|\theta - \theta'\|, & \|\nabla_x \hat{\ell}(\theta; x) - \nabla_x \hat{\ell}(\theta; x')\|_* &\leq L_{xx} \|x - x'\|, \\
\|\nabla_\theta \hat{\ell}(\theta; x) - \nabla_\theta \hat{\ell}(\theta; x')\|_* &\leq L_{\theta x} \|x - x'\|, & \|\nabla_x \hat{\ell}(\theta; x) - \nabla_x \hat{\ell}(\theta'; x)\|_* &\leq L_{x\theta} \|\theta - \theta'\|.
\end{aligned}
\tag{28}
$$

Let $\|\cdot\|_*$ be the dual norm to $\|\cdot\|$; we abuse notation by using the same norm $\|\cdot\|$ on $\Theta$ and $\mathcal{X}$. Here we have proved the second condition of Assumption 1 holds true by Theorem 2, with $L_{xx} = \frac{2M}{\sigma^2}$. Therefore, if $\hat{\ell}$ satisfies the other three conditions, we could adopt a similar proof procedure for Theorem 2 in Sinha et al. (2018) to prove the convergence of Algorithm 1.

## A.5 PROOF OF THEOREM 3

*Proof.* By Eq. 16 we could get

$$
\sup_{S \in \mathcal{P}} \mathbb{E}_S[\ell(\theta; s)] \leq \inf_{\gamma \geq 0} \{\gamma\rho + \mathbb{E}_{P_0}[\phi_\gamma(\theta; x_0)]\},
\tag{29}
$$

where

$$
\mathbb{E}_{P_0}[\phi_\gamma(\theta; x_0)] = \mathbb{E}_{P_0}\left\{\sup_{x \in \mathcal{X}} \mathbb{E}_Z[\ell(\theta; x + z) - \gamma c(x + z, x_0)]\right\}.
\tag{30}
$$

Since $\ell(\theta; x + z) - \gamma c(x + z, x_0)$ is strongly concave for $x + z$ in Sinha et al. (2018), by *Jensen Inequality* we have for any fixed $x$,

$$
\begin{aligned}
&\mathbb{E}_Z\{\ell(\theta; x + z) - \gamma c(x + z, x_0)\} \\
\leq &\ell(\theta; \mathbb{E}_Z(x + z)) - \gamma c(\mathbb{E}_Z(x + z), x_0) \\
= &\ell(\theta; x) - \gamma c(x, x_0).
\end{aligned}
\tag{31}
$$

Hence, the following inequality holds

$$
\mathbb{E}_{P_0}[\phi_\gamma(\theta; x_0)] \leq \mathbb{E}_{P_0} \sup_{x \in \mathcal{X}} [\ell(\theta; x) - \gamma c(x, x_0)].
\tag{32}
$$

By Proposition 1 in Sinha et al. (2018), we could get

$$
\inf_{\gamma \geq 0}\left\{\mathbb{E}_{P_0} \sup_{x \in \mathcal{X}} [\ell(\theta; x) - \gamma c(x, x_0)] + \gamma\rho\right\} = \sup_{P' : W_c(P', P_0) \leq \rho} \mathbb{E}_{P'}[\ell(\theta; x)].
\tag{33}
$$

Finally, we can get Eq. 7 by concatenating the inequalities which completes the proof. □

## A.6 CONNECTIONS BETWEEN ROBUSTNESS CERTIFICATES

**Proposition 1.** *Let $p_A, p_B$ denote the largest and second largest probabilities returned by the smoothed classifier $g(x_0)$ and $R = \frac{\sigma}{2}\left(\Phi^{-1}\left(\underline{p_A}\right) - \Phi^{-1}\left(\overline{p_B}\right)\right)$. We choose $\ell$ as the cross-entropy loss in the smoothed loss function $\hat{\ell}(x) = \mathbb{E}_Z[\ell(\theta; x + z)]$, $z \sim Z = \mathcal{N}(0, \sigma^2 I)$. If*

$$
\hat{\ell}(\theta; x) \leq -\log\left(\Phi\left[\Phi^{-1}(\overline{p_B}) + \frac{\|x - x_0\|_2}{\sigma}\right]\right)
\tag{34}
$$

*holds, and the ground truth label $y = c_A$, then $g(x)$ is robust against any $x$ such that $\|x - x_0\|_2 \leq R$.*

*Proof.* By Theorem 1 of Cohen et al. (2019), we just need to prove the condition Eq. 34 leads to the condition $\|x - x_0\|_2 \leq R$. With $\ell$ being the cross-entropy loss,

$$
\hat{\ell}(x) = \mathbb{E}_Z[\ell(\theta; x + z)] = \mathbb{E}_Z[-\log(f^{(y)}(x + z))].
\tag{35}
$$

Then we use *Jensen Inequality* on $-\log(x)$ to obtain

$$
\hat{\ell}(x) = \mathbb{E}_Z[-\log(f^{(y)}(x + z))] \geq -\log[\mathbb{E}_Z f^{(y)}(x + z)].
\tag{36}
$$

As $y = c_A$, we have $\mathbb{E}_Z f^{(y)}(x+z) = P(f(x+z) = c_A)$. By Eq. 34, we have

$$- \log[P(f(x+z) = c_A)] \leq \mathbb{E}_Z[-\log(f^{(y)}(x+z))] \leq -\log\left(\Phi\left[\Phi^{-1}(\overline{p_B}) + \frac{\|x - x_0\|_2}{\sigma}\right]\right).$$
(37)

And hence

$$P(f(x+z) = c_A) \geq \Phi\left[\Phi^{-1}(\overline{p_B}) + \frac{\|x - x_0\|_2}{\sigma}\right].$$
(38)

By the proof of Theorem 1 in Cohen et al. (2019),

$$P(f(x+z) = c_A) = \Phi\left(\Phi^{-1}\left(\underline{p_A}\right) - \frac{\|x - x_0\|_2}{\sigma}\right),$$
(39)

which leads to

$$\Phi^{-1}\left(\underline{p_A}\right) - \frac{\|x - x_0\|_2}{\sigma} \geq \Phi^{-1}(\overline{p_B}) + \frac{\|x - x_0\|_2}{\sigma}.$$
(40)

Therefore,

$$\|x - x_0\|_2 \leq \frac{\sigma}{2}\left(\Phi^{-1}\left(\underline{p_A}\right) - \Phi^{-1}\left(\overline{p_B}\right)\right) = R.$$
(41)

To sum up, if Eq. 34 holds and $x_0$ is correctly classified, $g(x)$ is robust within a $\ell_2$ ball with radius $R$. One can tell the loss on a single instance is weakly associated with the robustness of the model, and the condition of $g(x)$ being robust is quite stringent. It is not practical to sum up the single-instance loss to gauge the model robustness either. □

## B  EXPERIMENTS

### B.1  BASELINE SETTINGS

We provide the training settings for baselines in Table 3. The learning rate $\eta_1$ is adjusted according to different $\gamma$s and $\varepsilon$s. The noise level ($\sigma$) is the same for all methods.

| Dateset | mechanism | $\eta_1$ | $\eta_2$ | batch size | epochs |
|---|---|---|---|---|---|
| MNIST | WRM | $0.5/\gamma$ | $1 \times 10^{-4}$ | 128 | 25 |
| | STN | — | $1 \times 10^{-4}$ | 128 | 25 |
| | SmoothAdv | $\epsilon/2$ | $1 \times 10^{-4}$ | 128 | 25 |
| | TRADES | $\epsilon/2$ | $1 \times 10^{-4}$ | 128 | 25 |
| CIFAR-10 | WRM | $0.5/\gamma$ | $1 \times 10^{-4}$ | 128 | 100 |
| | STN | — | $1 \times 10^{-4}$ | 128 | 100 |
| | SmoothAdv | $\epsilon/2$ | $1 \times 10^{-4}$ | 128 | 100 |
| | TRADES | $\epsilon/2$ | $1 \times 10^{-4}$ | 128 | 100 |
| Tiny ImageNet | WRM | $0.5/\gamma$ | $2 \times 10^{-5}$ | 128 | 100 |
| | STN | — | $2 \times 10^{-5}$ | 128 | 100 |
| | SmoothAdv | $\epsilon/2$ | $2 \times 10^{-5}$ | 128 | 100 |
| | TRADES | $\epsilon/2$ | $2 \times 10^{-5}$ | 128 | 100 |

**Table 3:** Baseline hyperparameter settings. $\gamma$ and $\varepsilon$ is chosen from Table 1.

### B.2  RESULTS WITH VARYING $\sigma$ AND $(K, r)$

We compare NAL with SmoothAdv and STN under the same experimental setting but different $\sigma$s. In Table 4, NAL achieves the best performance at $\sigma = 0.1$ above all. We believe in different experimental settings, the best $\sigma$ value is different. For example, NAL and STN obtain the best performance at $\sigma = 0.1$, whereas SmoothAdv performs best at $\sigma = 0.05$. For the same $\sigma$, NAL has superior performance than the other two baselines except that, when $\sigma = 0.05$, SmoothAdv is more robust than NAL for $\ell_2$ attack radius $\geq 0.75$. This is mainly because SmoothAdv achieves the best performance when $\sigma = 0.05$. However, the model accuracies degrade below 0.5 is not our main consideration.

| $\ell_2$ attack radius | | 0 | 0.25 | 0.5 | 0.75 | 1 | 1.25 | 1.5 | 1.75 |
|---|---|---|---|---|---|---|---|---|---|
| NAL | $\sigma = 0.05$ | 0.8579 | 0.7809 | 0.6761 | 0.5549 | 0.4262 | 0.2916 | 0.1888 | 0.1329 |
| NAL | $\sigma = 0.1$ | 0.8522 | **0.8155** | **0.7684** | **0.7140** | **0.6466** | **0.5684** | **0.4829** | **0.3909** |
| NAL | $\sigma = 0.2$ | 0.8307 | 0.7781 | 0.7213 | 0.6498 | 0.5644 | 0.4785 | 0.3837 | 0.2959 |
| SmoothAdv | $\sigma = 0.05$ | 0.7643 | 0.7086 | 0.6378 | 0.5644 | 0.4841 | 0.4050 | 0.3297 | 0.2602 |
| SmoothAdv | $\sigma = 0.1$ | 0.8066 | 0.7264 | 0.6281 | 0.5376 | 0.4399 | 0.3467 | 0.2700 | 0.2010 |
| SmoothAdv | $\sigma = 0.2$ | 0.7411 | 0.6758 | 0.6079 | 0.5327 | 0.4689 | 0.4005 | 0.3350 | 0.2736 |
| STN | $\sigma = 0.05$ | **0.8988** | 0.7347 | 0.4834 | 0.2594 | 0.1167 | 0.0466 | 0.0155 | 0.0063 |
| STN | $\sigma = 0.1$ | 0.8669 | 0.7609 | 0.6164 | 0.4416 | 0.2847 | 0.1678 | 0.0927 | 0.0443 |
| STN | $\sigma = 0.2$ | 0.8000 | 0.7060 | 0.5867 | 0.4695 | 0.3523 | 0.2472 | 0.1708 | 0.1125 |

**Table 4:** Different methods with different levels of noise on CIFAR-10, ResNet-18, $\gamma = 1.5$ and $(K, r) = (4, 4)$. The best performance at the same noise level is in bold.

In Table 5, we show NAL's accuracy over a variety of $\sigma, K, r$ values. We found that the result of $\sigma = 0.12$ is generally better than a larger value. Under the same $\sigma$, we choose $K \in \{2, 4, 6, 8\}, r \in \{1, 4\}$,. We found the model cannot converge with $(K, r) = (2, 1)$, and thus did not present the results. The Table show that a larger $K$ admits better robustness whereas $r$ does not have that impact.

| $\ell_2$ attack radius | | | 0 | 0.25 | 0.5 | 0.75 | 1 | 1.25 | 1.5 | 1.75 |
|---|---|---|---|---|---|---|---|---|---|---|
| | $K$=2 | $r$=4 | **0.8593** | 0.8414 | 0.8152 | 0.7891 | 0.7584 | 0.7177 | 0.6699 | 0.6103 |
| | $K$=4 | $r$=1 | 0.8480 | 0.8142 | 0.7772 | 0.7306 | 0.6756 | 0.6185 | 0.5521 | 0.4748 |
| | $K$=4 | $r$=4 | 0.8462 | 0.8129 | 0.7728 | 0.7237 | 0.6692 | 0.6044 | 0.5339 | 0.4646 |
| $\sigma = 0.12$ | $K$=6 | $r$=1 | 0.8528 | 0.8312 | 0.8105 | 0.7803 | 0.7473 | 0.7081 | 0.6585 | 0.5989 |
| | $K$=6 | $r$=4 | 0.8424 | 0.7990 | 0.7584 | 0.7022 | 0.6372 | 0.5663 | 0.4853 | 0.3950 |
| | $K$=8 | $r$=1 | 0.8526 | **0.8418** | **0.8329** | **0.8194** | **0.8049** | **0.7883** | **0.7670** | **0.7365** |
| | $K$=8 | $r$=4 | 0.8443 | 0.8025 | 0.7494 | 0.6929 | 0.6250 | 0.5469 | 0.4578 | 0.3758 |
| | $K$=2 | $r$=1 | 0.7522 | 0.6885 | 0.6186 | 0.5403 | 0.4596 | 0.3739 | 0.2971 | 0.2195 |
| | $K$=2 | $r$=4 | 0.7953 | 0.7421 | 0.6801 | 0.6069 | 0.5226 | 0.4332 | 0.3498 | 0.2691 |
| | $K$=4 | $r$=1 | 0.7669 | 0.7077 | 0.6399 | 0.5717 | 0.4896 | 0.4103 | 0.3318 | 0.2594 |
| $\sigma = 0.25$ | $K$=4 | $r$=4 | 0.8121 | 0.7679 | **0.7143** | 0.6540 | 0.5882 | 0.5153 | 0.4381 | **0.3622** |
| | $K$=6 | $r$=1 | 0.7632 | 0.7082 | 0.6501 | 0.5790 | 0.5095 | 0.4349 | 0.3569 | 0.2805 |
| | $K$=6 | $r$=4 | 0.8098 | 0.7578 | 0.7059 | 0.6410 | 0.5677 | 0.4875 | 0.4133 | 0.3300 |
| | $K$=8 | $r$=1 | 0.7808 | 0.7285 | 0.6720 | 0.6073 | 0.5273 | 0.4490 | 0.3694 | 0.2951 |
| | $K$=8 | $r$=4 | **0.8150** | **0.7694** | 0.7132 | **0.6549** | **0.5896** | **0.5185** | **0.4391** | 0.3574 |
| | $K$=2 | $r$=1 | 0.6434 | 0.5899 | 0.5335 | 0.4739 | 0.4150 | 0.3524 | 0.2958 | 0.2422 |
| | $K$=2 | $r$=4 | 0.7122 | 0.6681 | 0.6201 | 0.5717 | 0.5212 | 0.4632 | 0.4090 | 0.3496 |
| | $K$=4 | $r$=1 | 0.6744 | 0.6165 | 0.5631 | 0.5048 | 0.4450 | 0.3829 | 0.3226 | 0.2652 |
| $\sigma = 0.5$ | $K$=4 | $r$=4 | 0.7186 | 0.6701 | 0.6217 | 0.5699 | 0.5135 | 0.4505 | 0.3966 | 0.3373 |
| | $K$=6 | $r$=1 | 0.6860 | 0.6335 | 0.5799 | 0.5154 | 0.4566 | 0.3979 | 0.3362 | 0.2811 |
| | $K$=6 | $r$=4 | 0.7185 | 0.6771 | 0.6243 | 0.5707 | 0.5174 | 0.4608 | 0.4019 | 0.3455 |
| | $K$=8 | $r$=1 | 0.6943 | 0.6424 | 0.5911 | 0.5380 | 0.4758 | 0.4181 | 0.3548 | 0.2968 |
| | $K$=8 | $r$=4 | **0.7239** | **0.6804** | **0.6320** | **0.5836** | **0.5345** | **0.4736** | **0.4250** | **0.3716** |

**Table 5:** NAL with different $\sigma$s and $(K, r)$ on CIFAR-10, ResNet-18 when $\gamma = 1.16$. The best performance under the same noise level is in bold.

| $\ell_2$ attack radius | 0 | 0.25 | 0.5 | 0.75 | 1 | 1.25 | 1.5 | 1.75 |
|---|---|---|---|---|---|---|---|---|
| ELU Model | 0.8596 | 0.8046 | 0.7348 | 0.647 | 0.5465 | 0.4387 | 0.3315 | 0.2351 |
| ReLU Model | 0.8522 | 0.8155 | 0.7684 | 0.714 | 0.6466 | 0.5684 | 0.4829 | 0.3909 |

**Table 6:** Testing accuracies for the ReLU model and the ELU model on CIFAR-10, ResNet-18.

## B.3 RESULTS WITH VARYING $\gamma$

We show the impact of $\gamma$ on MNIST and CIFAR-10. On MNIST, $\gamma$ takes the value $\{0.25, 1.5, 3\}$ and $\sigma$ is chosen as 0.05. On CIFAR-10, $\gamma \in \{0.25, 1.5, 5\}$ and $\sigma$ is set to 0.1. $(K, r) = (4, 4)$ for all experiments. Fig. 4(a) and 5(a) compare NAL with WRM on models with ELU, whereas the rest of Fig. 4 and 5 show the comparison with SmoothAdv, TRADES, and STN on regular models. NAL has superior performance than baselines in almost all cases.

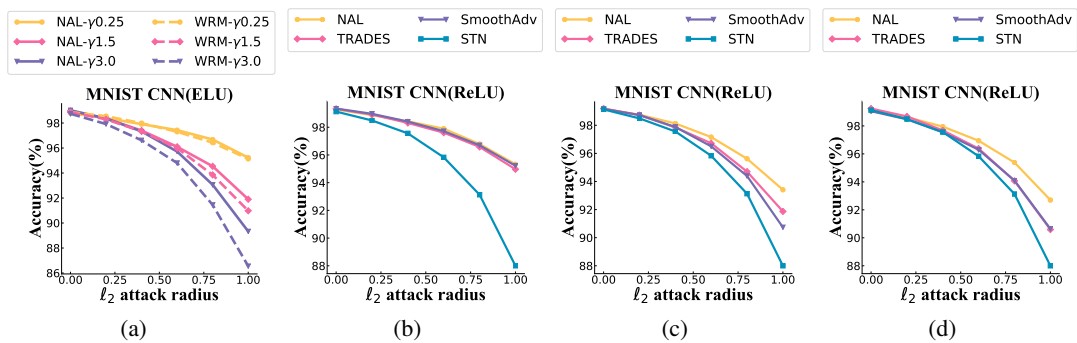

**Figure 4:** NAL versus baselines on MNIST, CNN. (a) NAL versus WRM for different $\gamma$s. (b) $\gamma = 0.25$. (c) $\gamma = 1.5$. (d) $\gamma = 3$. Equivalent $\varepsilon$s are used in SmoothAdv and TRADES.

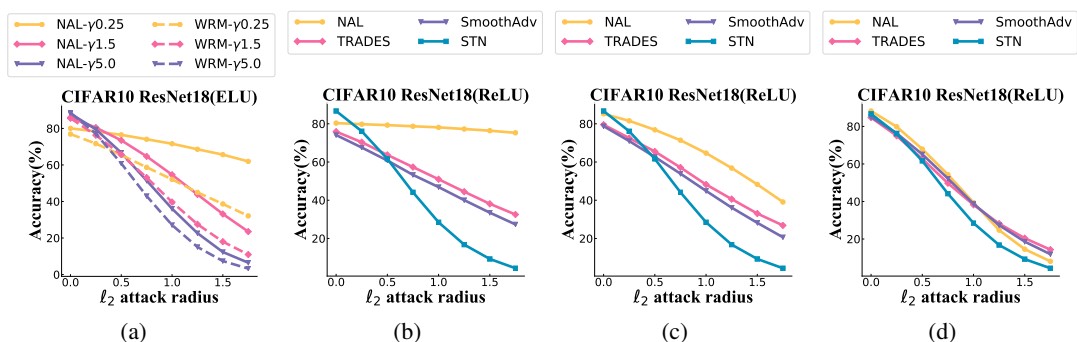

**Figure 5:** NAL versus baselines on CIFAR-10, ResNet-18. (a) NAL versus WRM for different $\gamma$s. (b) $\gamma = 0.25$. (c) $\gamma = 1.5$. (d) $\gamma = 5$. Equivalent $\varepsilon$s are used in SmoothAdv and TRADES.

The results on MNIST are similar to those obtained in the experiment section. The gaps between NAL and baselines in Fig. 4(c)(d) grow larger than that in 4(b), which could be explained different values of $\gamma$. On CIFAR-10, since we switch to ResNet-18, the experimental results are slightly different from that on VGG-16. The best robustness occurs at $\gamma = 0.25$ (Fig. 5(b)). The performance of STN is consistent with that in the experiment section: it enjoys the highest clean accuracies but much worse performance over adversarial examples. As we observe on Fig. 5(b),(c),(d), with an increase of $\gamma$, the clean accuracy increases but at the sacrifice of adversarial accuracies. For example, NAL degrades to the performance of SmoothAdv and TRADES at $\gamma = 5$.

### B.4 COMPARISON BETWEEN RELU AND ELU

Here we show the difference between ResNet-18 with ReLU and ELU on CIFAR-10 for NAL. From Fig. 6, throughout the training process, the loss of the ReLU model is smaller than that of the ELU

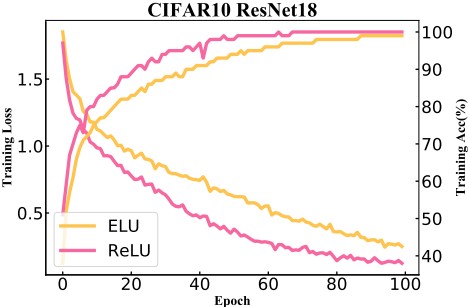

**Figure 6:** The comparison between the ReLU model (pink) and the ELU model (yellow) on CIFAR-10, ResNet-18 with $\gamma = 1.5$ and $\sigma = 0.1$. The ReLU model converges faster than the ELU model.

model, and ReLU model presents faster convergence. The robustness performance of both models is presented in Table 6. It is clear that in the testing phase, the ReLU model also obtains a better performance. Hence NAL generally yields better performance on ReLU models than ELU models.