# OpenReview forum: "Certified Distributional Robustness via Smoothed Classifiers "
_ICLR.cc/2021/Conference — Reject_

### Official Review · AnonReviewer2 · 2020-10-28
**A randomized smoothing distributional robust certificate**

**Rating:** 2
**Confidence:** 4

**Review:**

This paper studies the problem of certified robustness in adversarial learning. In a nutshell, they apply the randomized smoothing technique to the distributional robustness certificate proposed by Sinha et al. (2018), thereby relaxing the smoothness assumption required therein so that the ReLU network can be applied. Based on this new formulation, they derive the upper bound on the worst-case population loss and develop an algorithm with convergence guarantees. The results on tested on MNIST, CIFAR-10 and Tiny ImageNet.

The topic is definitely important and the authors did a good job of explaining their framework. Nevertheless, unfortunately, I think the proposed methodology seems rather straightforward and does not provide many new insights into this area. Besides, there are numerous careless flaws in the paper.

1. Theorem 1 appears to be a standard result in distributional robust optimization and it is unfortunate that the authors did not recognize it. See, for example, Kuhn et al. (2019) (https://arxiv.org/abs/1908.08729).

2. Theorem 2 and Corollary 1 appear to be a standard result in randomized smoothing. Besides, the proof is not rigorous -- it takes for granted that the $\hat\ell$ is twice differentiable.

3. The statement of Theorem 3 contains an obvious typo. The left side and the right side are no different from each other -- the only difference is the integration variable: one is $\hat{x}$ and the other is $x'$.

That is to say, all three major theoretical results are either straightforward corollary from existing results or contain flaws.

4. The algorithm proposed in Section 3.2 is problematic. In particular, in equation (6) and Alg.1 line 3-6, when $\cal X$ is a normed vector space, the value of the inner supremum does not depend on $z_{ij}$ simply via a change of variable from $x+z_{ij}$ to $x$. I think the correct version should be
$$
  \frac{1}{n} \sum_{i=1}^n \sup_{x\in\cal X} [ \frac{1}{s}\sum_{j=1}^s \ell(\theta;x+z_{ij}) - \gamma c(x,x_0^i) ].
$$

5. As for the numerical experiment, it is unclear to me how $\gamma$ is chosen, and in particular, does such choice make the inner supremum of (6) a strongly concave problem?

Given these many issues that are not easy to fix, I would encourage the authors to carefully revise their manuscript and resubmit to another conference.

---

> ### Author Response · Authors · 2020-11-22
> **Thanks for the valuable feedback. We will carefully address the concerns one by one.**
>
>
> Q1: Theorem 1 appears to be a standard result in distributional robust optimization and it is unfortunate that the authors did not recognize it. See, for example, Kuhn et al. (2019) (https://arxiv.org/abs/1908.08729).
>
> R1: The weak Kantorovich duality in Kuhn et al. (2019) can be summarized as
> $$\inf\_{\gamma \geq 0} \left\\{  \mathbb{E}\_{Q} \sup\_{x \in \mathcal{X}} \left[\ell(\theta ; x)-\gamma c \left(x, x_{0}\right) \right] + \gamma \rho \right\\} = \sup\_{x\sim P: W_{c}(P, Q) \leq \rho} \mathbb{E}\_{P}[\ell(\theta ; x)].$$
> However, it is not discussed that the certified robustness bound holds after introducing a randomized Gaussian variable $z.$ Our result is different from Kuhn et al. (2019) such that we prove the certificate holds with inserted noise. As we show in Theorem 3 we provide better certificate with inserted noise.
>
> Q2: Theorem 2 and Corollary 1 appear to be a standard result in randomized smoothing. Besides, the proof is not rigorous -- it takes for granted that the $\hat\ell$ is twice differentiable.
>
> R2:After adding noise, the $ \hat{\ell} $ function is infinitely differentiable, which we have explained in the revisions.
>
> Q3: The statement of Theorem 3 contains an obvious typo. The left side and the right side are no different from each other -- the only difference is the integration variable: one is $\hat{x}$ and the other is $x'$.
> That is to say, all three major theoretical results are either straightforward corollary from existing results or contain flaws.
>
> R3: The left and right sides of Theorem 3 are different. In $\hat{x}= x+z$, $x$ is the adversarial example, and $z \sim \mathcal{N}(0,\sigma^2) $. But $x^{\prime}$ on the right is the adversarial example from the result of Sinha et al. (2017). We compare the our loss function with that of Sinha et al. (2017) to show that our loss is smaller, and hence the robustness level is higher.
>
> Q4: The algorithm proposed in Section 3.2 is problematic. In particular, in equation (6) and Alg.1 line 3-6, when $\cal X$ is a normed vector space, the value of the inner supremum does not depend on $z_{ij}$ simply via a change of variable from $x+z_{ij}$ to $x$. I think the correct version should be $$ \frac{1}{n} \sum_{i=1}^n \sup_{x\in\cal X} [ \frac{1}{s}\sum_{j=1}^s \ell(\theta;x+z_{ij}) - \gamma c(x,x_0^i) ]. $$
>
> R4: It is not true in saying that the value of the inner supremum does not depend on $z_{ij}$. The algorithm is correct except for the objective should be $$ \frac{1}{n} \sum_{i=1}^n \sup_{x\in\cal X} [ \frac{1}{s}\sum_{j=1}^s \ell(\theta;x+z_{ij}) - \gamma c(x+z_{ij},x_0^i) ]. $$ We have made the correction in the revision.
>
>
>
> Q5: As for the numerical experiment, it is unclear to me how $\gamma$ is chosen, and in particular, does such choice make the inner supremum of (6) a strongly concave problem?
>
> R5: First, in Corollary 1, we show $\gamma \geq 2M $ as a condition for the loss to be strongly concave. And as we mention in the Experiment that three different values of $\gamma $ are used to test the model with different perturbation ranges, which are listed in Table 1.
>
>
> Aman Sinha and Hongseok Namkoong and John Duchi. Certifiable Distributional Robustness with Principled Adversarial Training. International Conference on Learning Representations, 2018.

---

### Official Review · AnonReviewer3 · 2020-10-29
**Concerning issues of novelty**

**Rating:** 2
**Confidence:** 5

**Review:**

The authors use some of the theory from optimal transport to certify the degree of robustness of an image classifier to adversarial perturbation. The theoretical contributions of the work, however, are largely already present, or easily deduced from results that are either already published or available online in a pre-published format. For theorem I have given a brief overview.  My score reflects a strong concern over the degree of originality of the results presented, particularly considering the authors cite some of the papers in which these results can already be found. I welcome the authors and other reviewers to draw my attention to any contributions in the paper I may have overlooked by mistake.

### Theorem 1
In essence Theorem 1 corresponds to weak Kantorovich duality for the robust optimal transport problem. The best reference for strong Kantorovich duality in this setting is [Blanchet & Murthy (2019)](https://doi.org/10.1287/moor.2018.0936) and the result is easily deduced from Section 2.2.1. More closely the result can also be deduced from Proposition 1 of [Sinha et al. (2017)](https://arxiv.org/abs/1710.10571),  [Kuhn et al. (2019)](https://pubsonline.informs.org/doi/10.1287/educ.2019.0198) also have a variety of results on this topic. Theorem 1 of  [Cranko et al. (2020)](https://arxiv.org/pdf/2002.04197.pdf) additionally qualifies the tightness of the upper bound and relaxes a number of assumptions of some of the aforementioned papers. There are a number of other, related papers by some of the authors mentioned that I have left out.

### Theorem 2
Theorem 2 is not particularly surprising. In the proof the authors implicitly assume $\ell(\theta,\cdot)$ is twice differentiable and bounded (for all $\theta$). The addition of Gaussian noise is an additional complication, but it does not change things too much. The theorem is unsurprising because if it were not true it would lead to an absurd contradiction with the boundedness assumption.  The strong concavity result in the following corollary is also observed by [Sinha et al. (2017)](https://arxiv.org/abs/1710.10571) (at the bottom of page 2).

### Theorem 3
Theorem 3 was already proven in a significantly more general setting by [Blanchet & Murthy (2019)](https://doi.org/10.1287/moor.2018.0936)  (Theorem 1). Furthermore I am not convinced of the correctness of the proof since there appears to be an assumption that $\gamma$ is sufficiently larger for $\ell(\theta; x) - \gamma c(x,x_0)$ to be strongly concave. And I cannot even find an assumption that at this point $c(\cdot, x_0)$  is assumed so much as convex for a choice of $x_0\in\mathcal{X}$.

---

> ### Author Response · Authors · 2020-11-22
> **Thanks for the valuable feedback. We will carefully address the concerns one by one.**
>
> Q1: Theorem 1
>
> In essence Theorem 1 corresponds to weak Kantorovich duality for the robust optimal transport problem. The best reference for strong Kantorovich duality in this setting is Blanchet & Murthy (2019) and the result is easily deduced from Section 2.2.1. More closely the result can also be deduced from Proposition 1 of Sinha et al. (2017), Kuhn et al. (2019) also have a variety of results on this topic. Theorem 1 of Cranko et al. (2020) additionally qualifies the tightness of the upper bound and relaxes a number of assumptions of some of the aforementioned papers. There are a number of other, related papers by some of the authors mentioned that I have left out.
>
> R1: It is not mentioned in those works what the certificate is with randomized noise. We provide a certificate in this case, and show the result can be readily deployed to large-scale dataset. We show in detail how our result is different from those works: The weak Kantorovich duality in Blanchet & Murthy (2019), Sinha et al. (2017), Kuhn et al. (2019) can be summarized as
>
> $$ \inf\_{\\gamma \\geq 0} \left\\{  \mathbb{E}\_{Q} \sup\_{x \\in \\mathcal{X}} \left[\ell(\theta ; x)-\gamma c \left(x, x_{0}\right) \right] + \gamma \rho \right\\} = \sup\_{x\sim P: W_\{c}(P, Q) \leq \rho} \mathbb{E}\_{P}[\ell(\theta ; x)].$$
> However, it is not discussed that the certified robustness bound holds after introducing a randomized Gaussian variable $z.$ Theorem 1 in our paper improves upon the weak Kantorovich duality to provide a robustness certificate with the additive Gaussian noise.
>
> The author would like to have the paper compared with the series of randomized smoothing works such as Lecuyer et al. (2019), Li et al. (2019), Cohen et al.(2019), etc. We give a more convincing robustness guarantee by ensuring the perturbed example to be correctly classified, rather than being classified to the same label as the original example (which can be wrong).
>
>
> Q2: Theorem 2
>
> Theorem 2 is not particularly surprising. In the proof the authors implicitly assume $\ell(\theta,\cdot)$ is twice differentiable and bounded (for all $\theta$). The addition of Gaussian noise is an additional complication, but it does not change things too much. The theorem is unsurprising because if it were not true it would lead to an absurd contradiction with the boundedness assumption. The strong concavity result in the following corollary is also observed by Sinha et al. (2017) (at the bottom of page 2).
>
> R2: It is not true in saying that the addition of noise does not change things much. Above all, it makes \hat{\ell} have derivatives of all orders as shown in our revisions. Second, Theorem 2 proves the smoothness of the loss function $ \hat{\ell}(\theta; x) :=  \mathbb{E}_{z} [\ell(\theta ; x+z)],~z \sim \mathcal{N}(0,\sigma^2I) $. What we assume is that the function $\ell (\theta, \cdot) $ is bounded. It should be noted that the two functions $ \hat{\ell} $ and $\ell $ are different. Theorem 2 cannot be directly deduced from Sinha et al. (2017) due to the inclusion of randomized noise in the loss function.
>
> Q3: Theorem 3
>
> Theorem 3 was already proven in a significantly more general setting by Blanchet & Murthy (2019) (Theorem 1). Furthermore, I am not convinced of the correctness of the proof since there appears to be an assumption that $\gamma$ is sufficiently larger for $\ell(\theta; x) - \gamma c(x,x_0)$ to be strongly concave. And I cannot even find an assumption that at this point $c(\cdot, x_0)$ is assumed so much as convex for a choice of $x_0 \in \mathcal{X}$.
>
> R3: First, the authors do not think that this is a special form of Blanchet &amp; Murthy (2019) (Theorem1), because Theorem 3 compares two different supremums, which does not appear in Blanchet &amp; Murthy (2019) (Theorem1). Second, \gamma is a design choice which we show in Corollary 1 that $\gamma \geq 2M $ for the loss to be strongly concave. Similar design choice is also made in Sinha et al. (2017). Third, $c(\cdot, x_0)$ is also a design choice to be a convex function to give the robustness certificate.
>
> Bai Li, Changyou Chen, Wenlin Wang, and Lawrence Carin. Certified adversarial robustness with additive noise. In Advances in Neural Information Processing Systems, pp. 9464–9474, 2019.
>
> Mathias Lecuyer, Vaggelis Atlidakis, Roxana Geambasu, Daniel Hsu, and Suman Jana. Certified robustness to adversarial examples with differential privacy. In 2019 IEEE Symposium on Security and Privacy (SP), pp. 656–672. IEEE, 2019.
>
> Cohen, Jeremy, Elan Rosenfeld, and Zico Kolter. Certified Adversarial Robustness via Randomized Smoothing. In International Conference on Machine Learning, pp. 1310-1320. 2019.
>
> Aman Sinha and Hongseok Namkoong and John Duchi. Certifiable Distributional Robustness with Principled Adversarial Training. International Conference on Learning Representations, 2018.

---

### Official Review · AnonReviewer1 · 2020-10-29

**Rating:** 3
**Confidence:** 4

**Review:**

This paper proposes smoothing the classifier in the distributional robust learning framework by adding random noise to the input. The smoothed distributional robust framework is used to gain robustness against adversarial perturbations in settings where the classifier is originally non-smooth and then smoothed via the additive noise. While the proposed idea can be potentially useful for training adversarially-robust classifiers over non-smooth function spaces, the paper's theoretical formulation seems to reduce to original non-smoothed distributional robust optimization. Theorem 2 also seems incorrect and its proof suffers from several mistakes.

To see the main issue with the paper's problem formulation, note that Theorem 1 results in the adversarial loss function \phi_{\gamma}= E_Z[ max_x{...} ] which first maximizes over x\in \mathcal{X} (inner operation) and then takes expectation over Z (outer operation). This sequence is also the case in Equation (6) and is the basis of the theoretical and numerical analysis throughout the paper. However, one can see that this objective E_Z[ max_x{...} ] simply reduces to max_x{...} with no expectation over Z. This is because as long as the support set \mathcal{X} is unconstrained, which is the case in Algorithm 1 applying no projections on X, the maximization solution for x+z will be the same given any outcome Z=z. This is a direct consequence of maximizing a strongly-concave objective in \phi_{\gamma} that has a unique maximizer. In fact, I think the opposite order max_x{ E_Z[...] } which hasn't been analyzed in the paper will lead to a properly smoothed optimization problem. I, therefore, suggest redoing the analysis for the properly-ordered max_x{ E_Z[...] }.

Furthermore, Theorem 2 seems incorrect and the constant 2M in the theorem's upper-bound should be replaced with a function of both M and \sigma. In the theorem's proof, Equation (17) implicitly assumes that \sigma=1, whereas this assumption has not been mentioned in the theorem. Also, the step from Equation (22) to (23) mistakenly substitutes 1 with M and obtains a constant 2M instead of M+1, while there are no assumptions on M>1. The theorem needs to be revised since the current version is incorrect. Because of the above issues, I don't recommend this paper for acceptance in its current form. The paper should be revised to resolve these major issues.

---

> ### Author Response · Authors · 2020-11-22
> **Thanks for the valuable feedback. We will carefully address the concerns one by one.**
>
> Q1: To see the main issue with the paper's problem formulation, note that Theorem 1 results in the adversarial loss function \phi_{\gamma}= E_Z[ max_x{...} ] which first maximizes over x\in \mathcal{X} (inner operation) and then takes expectation over Z (outer operation). This sequence is also the case in Equation (6) and is the basis of the theoretical and numerical analysis throughout the paper. However, one can see that this objective E_Z[ max_x{...} ] simply reduces to max_x{...} with no expectation over Z. This is because as long as the support set \mathcal{X} is unconstrained, which is the case in Algorithm 1 applying no projections on X, the maximization solution for x+z will be the same given any outcome Z=z. This is a direct consequence of maximizing a strongly-concave objective in \phi_{\gamma} that has a unique maximizer. In fact, I think the opposite order max_x{ E_Z[...] } which hasn't been analyzed in the paper will lead to a properly smoothed optimization problem. I, therefore, suggest redoing the analysis for the properly-ordered max_x{ E_Z[...] }.
>
> R1: Thanks for pointing it out. We have corrected the mathematical expression as max_ x{ E_ Z[...] }. Actually, Algorithm 1 in the paper and our original implementation follows max_ x{ E_ Z[...] }. The experimental results should be considered valid.
>
>
> Q2: Furthermore, Theorem 2 seems incorrect and the constant 2M in the theorem's upper-bound should be replaced with a function of both M and \sigma. In the theorem's proof, Equation (17) implicitly assumes that \sigma=1, whereas this assumption has not been mentioned in the theorem. Also, the step from Equation (22) to (23) mistakenly substitutes 1 with M and obtains a constant 2M instead of M+1, while there are no assumptions on M>1. The theorem needs to be revised since the current version is incorrect. Because of the above issues, I don't recommend this paper for acceptance in its current form. The paper should be revised to resolve these major issues.
>
> R2: We have corrected Theorem 2 as well as its proof by which we can still guarantee \hat{l} to be smooth but with a different constant.

---

### Official Review · AnonReviewer4 · 2020-11-08
**Promising idea, writing needs work**

**Rating:** 6
**Confidence:** 3

**Review:**

##########################################################################

Summary: The authors propose to use the worst-case population loss (with bounded Wasserstein distance) over noisy inputs as a robustness metric. They provide a tractable upper
bound serving as a robustness certificate by exploiting the duality. The smoothness
of the loss function ensures the problem easy to optimize even for non-smooth
neural networks (since these are smoothed by expectation over noisy inputs). They show experiments on a variety of datasets and models to verify that in terms of empirical accuracies, their approach exceeds the state-of-the-art certified/heuristic methods in defending adversarial examples.


##########################################################################

Reasons for score:

Though the problem setting is not very accurately motivated, I liked the fact that the authors showed distribution dependent guarantees on robustness. It is still unclear to me though why adversarial noise is expected to be within some bounded Wasserstein distance from the probability distribution of the data. Also, their method seems incremental over Sinha et. al's work. Computational experiments look promising.

##########################################################################

Pros:

- the authors show a concave upper bound on the loss function and minimize that,
- they show that their method of "noisy adversarial learning" (NAL) has a convergence rate O(1/\sqrt{T}), which is similar to Sinha et al. (2018), but NAL does not need to replace the non-smooth layer ReLU with Sigmoid or ELU to guarantee robustness.
- improved performance in experiments

##########################################################################

Cons:

1. The authors say that "It is found that when fed with the perturbed instance x (within a L2 ball centered at x_0), a smoothed classifier g(x) = E_z[f(x + z)] with z \sim N (0, \sigma^2 I) can provably return the same label as $g(x_0)$ does.  However, we think such a robustness guarantee cannot ensure g(x0) to be correctly classified as y, resulting in unsatisfying performance in practice." I think there is a language error in this description, but if I understand correctly what the authors are trying to say, the smoothed classifiers are unsatisfying in performance -- however, aren't adversarial perturbations very specific, and therefore, smoothing should actually give more robustness to the model?

2. "Instead, we evaluate robustness as the worst case loss over the distribution of noisy inputs" - wouldn't the worst-case distribution simply place all the probability mass over the most adversarial example in the neighborhood?

3. "our approach does not require \ell to be smooth, and thus can be applied to
arbitrary neural networks." -- isn't the smoothness simply coming from the expected loss over distributions "close" to the input distribution? (This is clear later, since they bound the "possible probability distributions" by Wasserstein distances, but very confusing in the introduction.)

4. How loose/tight is the upper bound in Theorem 3? If \gamma*p is large, then the guarantee is almost meaningless.

##########################################################################

Questions during rebuttal period:


Please address and clarify the cons above


#########################################################################

Typos:

"The smoothness of the loss function ensures the problem easy to optimize even for non-smooth
neural networks."  -- grammar

The authors use "P" in the introduction before defining it. \mathcal{P} should be defined as the set of distributions at a bounded distance from P_0!!

Grammatical error: "But such a question remains open in the randomized regime, where randomized
smoothing can be considered as a contributing effort"

"Our work view the robustness" --> "views"

"right-hand side take the expectation" --> "takes"

---

> ### Author Response · Authors · 2020-11-22
> **Thanks for the valuable feedback. We will carefully address the concerns one by one.**
>
> Cons:
>
> Q1: The authors say that "It is found that when fed with the perturbed instance x (within an L2 ball centered at x_0), a smoothed classifier g(x) = E_z[f(x + z)] with z \sim N (0, \sigma^2 I) can provably return the same label as g(x0) does. However, we think such a robustness guarantee cannot ensure g(x0) to be correctly classified as y, resulting in unsatisfying performance in practice." I think there is a language error in this description, but if I understand correctly what the authors are trying to say, the smoothed classifiers are unsatisfying in performance -- however, aren't adversarial perturbations very specific, and therefore, smoothing should actually give more robustness to the model?
>
> R1: What we are trying to say is that the conventional smoothing methods do not necessarily increase robustness since the classification of the original input may not be correct. Robustness should be associated with a model, rather than a specific input example.
>
> Q2: "Instead, we evaluate robustness as the worst case loss over the distribution of noisy inputs" - wouldn't the worst-case distribution simply place all the probability mass over the most adversarial example in the neighborhood?
>
>
> R2: It is not true. The worst-case adversarial example is typically found by adding specific perturbations to an input sample. Those adversarial examples may be surrounded by many correctly classified examples, i.e., those adversarial examples can easily disappear if the decision boundary swings a little. Those adversarial examples are what we give up, and we instead focus on the adversarial examples of which their surroundings are also adversarial, forming a region that a classifier cannot easily escape. This is the reason that we consider the worst-case loss over distribution rather than a single instance.
>
>
> Q4: How loose/tight is the upper bound in Theorem 3? If \gamma*p is large, then the guarantee is almost meaningless.
>
> R4: Theorem 3 is not meant to show how tight the upper bound is, but merely to display that our method achieves better robustness. It is required that \rho has to be small for the certificate to hold, and thus \gamma*\rho cannot be large.

---

### Decision · Program_Chairs · 2021-01-07
**Final Decision**

**Decision:**

Reject

**Comment:**

The authors present a framework for deriving distributional robustness certificates for smoothed classifiers under perturbations of the input distribution bounded under the Wasserstein metric.

Several authors raised concerns regarding the correctness of results presented in the initial version of the paper. While these were addressed during the rebuttal, the reviewers remain concerned about the novelty of the work relative to prior work, in particular the following papers:
https://arxiv.org/abs/1908.08729
https://arxiv.org/pdf/2002.04197.pdf
https://doi.org/10.1287/moor.2018.0936
and the author responses during the rebuttal did not sufficiently address these concerns.

Hence, I recommend rejection.